# Modeling Optimal Location Distribution for Deployment of Flying Base Stations as On-Demand Connectivity Enablers in Real-World Scenarios

**DOI:** 10.3390/s21165580

**Published:** 2021-08-19

**Authors:** Jiri Pokorny, Pavel Seda, Milos Seda, Jiri Hosek

**Affiliations:** 1Department of Telecommunications, Faculty of Electrical Engineering and Communication, Brno University of Technology, Technicka 12, 616 00 Brno, Czech Republic; sedap@vut.cz (P.S.); hosek@feec.vutbr.cz (J.H.); 2Unit of Electrical Engineering, Tampere University, Korkeakoulunkatu 7, 337 20 Tampere, Finland; 3Institute of Automation and Computer Science, Brno University of Technology, Technicka 2, 616 69 Brno, Czech Republic; seda@fme.vutbr.cz

**Keywords:** UAV base station, flying base station, FBS, location optimization, network coverage capacity, on-demand, location covering problem, 5G

## Abstract

The amount of internet traffic generated during mass public events is significantly growing in a way that requires methods to increase the overall performance of the wireless network service. Recently, legacy methods in form of mobile cell sites, frequently called cells on wheels, were used. However, modern technologies are allowing the use of unmanned aerial vehicles (UAV) as a platform for network service extension instead of ground-based techniques. This results in the development of flying base stations (FBS) where the number of deployed FBSs depends on the demanded network capacity and specific user requirements. Large-scale events, such as outdoor music festivals or sporting competitions, requiring deployment of more than one FBS need a method to optimally distribute these aerial vehicles to achieve high capacity and minimize the cost. In this paper, we present a mathematical model for FBS deployment in large-scale scenarios. The model is based on a location set covering problem and the goal is to minimize the number of FBSs by finding their optimal locations. It is restricted by users’ throughput requirements and FBSs’ available throughput, also, all users that require connectivity must be served. Two meta-heuristic algorithms (cuckoo search and differential evolution) were implemented and verified on a real example of a music festival scenario. The results show that both algorithms are capable of finding a solution. The major difference is in the performance where differential evolution solves the problem six to eight times faster, thus it is more suitable for repetitive calculation. The obtained results can be used in commercial scenarios similar to the one used in this paper where providing sufficient connectivity is crucial for good user experience. The designed algorithms will serve for the network infrastructure design and for assessing the costs and feasibility of the use-case.

## 1. Introduction

Internet ubiquity has become natural in the modern world and the demand for it keeps growing significantly. People use the internet for social networking, streaming multimedia data, playing games, work, and many other things. However, in some cases, the demand exceeds the offering and users are not provided with enough throughput for sharing their data. This can be caused by obsolete telecommunication infrastructure or when user demands exceed, by multiple times, the infrastructure capabilities. Such a situation is typical, especially during large-scale events, where the data demand is temporarily raised above the infrastructure limits. During such events, implementation of supporting network infrastructure is mandatory to satisfy user requirements.

Recently, to cope with this imbalance, there were a few solutions introduced as, e.g., Cell on Wheels (COW) or portable base station that was brought to the affected area. However, all those technologies are limited especially in terms of deployment speed and operational costs. Therefore, one of the alternative solutions can be the utilization of unmanned aerial vehicles and their availability enabled through a rapid development of modern technologies. The unmanned aerial vehicles can be applied in various sectors like patrolling, delivery, video recording, and also as on-demand connectivity providers. In fact, there are already many commercial and research concepts where unmanned aerial vehicles are used as portable base stations. Unmanned Aerial Vehicle (UAV) base station, or Flying Base Station (FBS) can benefit from the most advantageous features of UAV, e.g., fast deployment time, mobility, and low cost.

When it comes to size, COWs are at least ten to twenty times bigger and heavier, or possibly even more, than a high volume FBS. That also means that the FBS can provide a lower data-rate than a COW. In large-scale scenarios, it is then very likely to require more than a single FBS. The distribution of users can be random or clustered into groups of different sizes. Each user can require different data throughput. This raises the question of how to optimally distribute multiple FBSs over a large area to provide the required data-rate to all users.

The described research follows our previous work [1], where we made a proof of the concept of the directional backhaul link for purposes of network throughput improvements in dense areas. In our work, the FBSs are used for assisting current infrastructure to extend network throughput for on-demand scenarios. The idea is demonstrated in Figure 1. There is an event with large amount of users that requires much higher network throughput than the current infrastructure can provide. The number of FBSs is determined from the users’ demand and from the available throughput from the local infrastructure. The problem described in this work is a type of coverage problem.

This paper presents a mathematical model for FBS distribution over an area. User demands and FBS throughput capacities are utilized as restricting aspects. The model is verified on a realistic music festival scenario. Heuristic algorithms were used to implement the model since its computational complexity, which is derived from the Set Covering Problem (SCP) problem is at least $O(n^{2})$. It is known from the no free lunch theorem [2] that no heuristic can be considered as better than others for all the problems and their datasets. However, in the recent research on the problem of set covering-based topics the differential evolution algorithm seems promising [3]. The other promising algorithm from the available meta-heuristic algorithms is the cuckoo search that is widely used in recent literature for a wide area of optimization problems as is: (i) forest cover classification [4], (ii) load balanced data gathering [5], (iii) permutation flow shop scheduling problem [6], and many others [7,8]. For that reason, the cuckoo search and differential evolution algorithms were used including the custom modification that contains the repairOperator (see Algorithm 3), to provide an efficient FBS placement.

The main contributions of this paper are as follows:Design of a novel model for FBS distribution over a selected area: This model is derived from SCP. Due to the high demand for data-rates, four main restricting aspects are considered, (i) user and base station capacities (for both downlink and uplink), (ii) FBS backhaul link throughput, (iii) consideration of existing base station nodes in the area to cover, (iv) the possibility to select locations with lower priority in the given area. This model provides the minimum number of required FBSs and their optimal locations. This knowledge is to be used in commercial applications;Implementation of two modified heuristic algorithms: differential evolution and cuckoo search were used to obtain a solution for the designed model. Differential evolution is well suited for set covering-based problems. Cuckoo search is a more recent algorithm widely used in optimization problems. Algorithms can be set for obtaining results where all users are provided with the internet coverage or the percentage of all users in case the number of FBS exceeds the maximum available limit;Verification of the model on real life scenario: overall feasibility of the two implemented algorithms was verified on a specific real-world scenario. Resulting number of FBSs and calculation time were used as the key performance identifiers.

## 2. Literature Review and State of the Art Discussion

FBSs can be utilized in a number of different use-cases, e.g., post-disaster, coverage/capacity support of local infrastructure, IoT data collection, etc. In all use-cases, the FBSs are used as an access point or relays for UEs on the ground. Depending on various parameters of the use-case, FBSs have different requirements to fulfill. Most of the research works in the topic of FBS location optimization have similar objectives with a common goal to either minimize or maximize the desired parameter in order to optimize the performance. The objectives can be summarized into two following areas:(i) maximization of UE coverage, power efficiency (endurance of UAVs), spectral efficiency, and (ii) minimization of the number of UAVs, and interferences. In our work, we focus on optimal UAV distribution over an area with the goal of minimizing the number of FBSs. This will lead to lower cost and complexity of the solution. This research can be used for both 2D and 3D FBS distribution. FBS trajectory optimization problems are not a part of the scope of this work, i.e., after FBSs are placed in the designated location, they continuously hover without moving to another location.

FBSs were discussed in numerous research works. Fotouhi et al. in [9] investigate a new mobility model for FBSs for improving the performance of cellular networks. The same authors propose, in [10], a mobility control algorithm to position FBSs to a better location to improve data throughput. In [11], Mignardi et al. propose a trajectory design of an FBS in order to improve the terrestrial base station performance. The number of studies concerning FBS increased rapidly from 2016. UAV location optimization is a problem that needs to be solved in any FBS use case. Coverage control problems of multiple UAV scenarios are discussed in [12], a review focusing on coverage methods for collective behavior of UAVs. Another review on location optimization problems was made by Cicek et al. [13] where the authors specifically target optimization methods for FBSs. To the best of author’s knowledge, these are the only two relevant overviews on this topic. According to the second overview, research studies can be divided into three main branches—static, semi-dynamic, and dynamic. Static is where UAVs and User Equipment (UE)s are stationary, semi-dynamic where UAVs can move freely but UEs are stationary, and dynamic, where UAVs and UEs can dynamically change their location. These can be further divided into scenarios with single or multiple UAVs. Our research focuses on a dynamic scenario with multiple UAVs.

UAV location can be optimized by means of different algorithms. Authors in [13] divided these algorithms into five groups: (i) exact—the algorithm is capable of finding the global optimum; (ii) well known heuristic algorithms, such as Dynamic Programming (DP), Particle Swarm Optimization (PSO), Genetic Algorithm (GA), or Gradient Algorithm (GDA); (iii) learning algorithms—these algorithms use learning procedures; (iv) enumeration—finding the best solution using exhaustive search; and (v) Problem Specific Heuristic (PSH)—a heuristic algorithm modified according to the problem properties. PSH algorithms are the most used from the list of studies, because they are most likely to give better results since they are always suited to a specific case. PSH algorithm is also used in our work, specifically Cuckoo Search (CUCKS) and Differential Evolution (DE) algorithms in modified version to serve our models.

DE is a heuristic algorithm that was developed in 1997 [14]. Differential evolution was used in many previous works for UAV path planning, e.g., in [15] the authors propose a UAV covering method with differential evolution as a cost optimization algorithm. A method for improving energy efficiency and optimize path planning was proposed in [16] and in [17]. CUCKS [18] is a relatively new algorithm introduced in 2009. It is a meta-heuristic algorithm inspired by cuckoo birds behavior. Three research works were found related to UAV distribution that used cuckoo search. In [19] the authors proposed an improved discreet cuckoo search algorithm for reconnaissance mission planning. Trajectory planning based on CUCKS was proposed in [20], where the authors focused on energy efficiency and throughput optimization. CUCKS was compared to particle swarm optimization in [21], the goal here was to evaluate online route planning methods.

In addition, the papers summarized in the overview [13], a summary of most recent papers is provided in this section, taking into account papers between the years 2018 and 2021. All papers focus on the location optimization problem with FBSs, i.e., how to optimally distribute the FBSs in order to minimize or maximize one or more parameters. Twelve papers from the past four years were selected and they are summarized in Table 1. The papers form four groups according to their goals. In the first group, the authors aim to minimize the number of required FBSs [22,23,24,25,26], in the second, to achieve maximum coverage of UEs [26,27,28,29,30], in the third, to maximize the network throughput [31,32], and, in the fourth, to maximize the spectral efficiency [33]. The optimization problem is either solved by existing algorithms or their combination [24,27,28,32,33] or a new algorithm is developed or derived from a previous algorithm [22,23,25,29,31].

In our research, the optimization problem is solved by the CUCKS and DE algorithms. Neither of the algorithms were used for the optimization similar to ours, i.e., minimization of the number of FBSs. The algorithms were selected as promising algorithms recently used for a wide area of optimization problems.

## 3. Design of Mathematical Model and Its Implementation

The effective deployment of UAV across a selected area is a difficult task. In this work, the enhancement of location covering models is presented (see Section 3.4) for the UAV deployment for on-demand connectivity scenarios. To ease the mathematical model readiness, in Table 2 we provide the terminology used in the remaining part of this paper adapted to the terms used in the literature.

### 3.1. Deployment Model

The design of our models is based on the so-called Location Set Covering Problem (LSCP) [34] and Maximal Covering Location Problem (MCLP) [35] models from the area of facility location problems. These models have many extensions presented in the literature, however, none of these models and extensions fit into our use-case. The gap in the literature that we encounter is that the models are not considering the combination of the following factors:(i) each demand capacity is assigned to just one facility at a given moment;(ii) consideration of existing services (the capacity of existing BTS nodes in a given area must be taken into account);(iii) the facility capacities and demand capacities should be represented separately for downlink and uplink and not as just a number altogether because the reserved ratio for uplink and downlink may differ for each node separately;(iv) the possibility to select locations from the original dataset that may not be covered. This is important when we find out that to cover the whole area we need more facilities (UAVs) than is available. We can reduce the less important areas and probably save some facilities.

Implicitly we assume that the requirement of the coverage availability (distance or sufficient signal power in our case including interference consideration) from facility to demand is always met. Further in this paragraph, we reference the above-mentioned model requirements when discussing the available optimization models in the literature. The mathematical model and the need for the first requirement (i) was originally discussed in [36,37] but this need was not defined in the model, only discussed. Further, in [38], the model included that requirement. The second requirement (ii) is mathematically defined in [39] for LSCP and further extended for MCLP model in [40]. The third aspect (iii) is, to the best of our knowledge, not covered in the literature, even the recent article [41] on that topic does not consider it. The last requirement (iv) is a special version of the so-called Multi-Service Location Set Covering Problem (MS-LSCP) that is considering multiple coverages of specific demands [42] but always set to zero since the coverage requirement from the demand for a facility is always zero. Based on that, we developed a new model that targets all the above-mentioned requirements altogether.

To derive a mathematical model let us set the following notation:*I* = a set of facility sites (UAV) 1,2,⋯,m;*J* = a set of demand areas (customers) 1,2,⋯,n;$d_{ij}$ = the shortest distance between facility *i* and demand *j*;$D_{max}$ = maximum distance which will be accepted for operation between the facilities and demands;$l_{j}$ = number of facilities required for servicing demand *j*;$x_{i}\in \left\{0,1\right\}$, where $x_{i}=1$ means that facility *i* is selected, while $x_{i}=0$ means that it is not selected;$N_{j}=\left\{i|d_{ij}\leq D_{max}\right\}$.

For the sake of simplicity, the facility sites will be referred as facility and demand areas as demand.

The MS-LSCP can be formulated as follows: minimize the number of facilities needed to cover the whole area, and locate them in such a manner to provide coverage for each demand by a requested number of facilities for a specific demand. In practice, this is important for back-up coverage for especially important demands to reduce the cases when some of the facilities fail and the important demand loses the connection. Formally:

Minimize
(1)$$\sum_{i\in I}x_{i}$$
subject to
(2)$$\forall j\in J:\;\sum_{i\in N_{j}}x_{i}\geq l_{j}$$
(3)$$\forall i\in I:\;x_{i}\in \left\{0,1\right\}$$

If $l_{j}=1$ for each demand *j*, then the above model is simplified to a single case, known as the classical LSCP. If $l_{j}=0$ then demand *j* does not need to be covered.

However, in real situations besides (or instead of) the MS-LSCP within the predefined range, it is more important to consider capacities of facilities. Since in our considered scenario we have to deal with the high density of users that are simultaneously connected, we assume the following additional notations:$C_{i}$ = capacity of facility *i*;$a_{j}$ = amount of demand at *j*;$y_{ij}\in \left\{0,1\right\}$ = non-fragmented demand from location *j* is assigned (1) or is not assigned (0) to facility *i*.

Further, there is a set of existing facilities $E_{f}$ (existing base station nodes in the area), and its corresponding decision variables $x_{i},i\in E_{f}$ are set to 1. Now, assume that capacities and demands are divided into uploads and downloads. For example, from the user’s point of view, the demand of 100 Mbit/s may be divided into 80 Mbit/sec for download and 20 Mbit/s for upload. To reach that expectations assume the following:$C_{i}^{u}$ = upload capacity of facility *i*;$C_{i}^{d}$ = download capacity of facility *i*;$a_{j}^{u}$ = upload amount of demand at *j*;$a_{j}^{d}$ = download amount of demand at *j*.

However, it still must be satisfied that the demand *j* (for download and upload at the same time) is directed to just one facility *i*, and the meaning of $y_{ij}$ remains the same as mentioned above. Now, let us combine all these assumptions into the following model:

Minimize
(4)$$\left(1+\varepsilon \right)\sum_{i\notin E_{f}}x_{i}+\sum_{i\in E_{f}}x_{i}$$
subject to
(5)$$\forall j\in J:\;\sum_{i\in N_{j}}x_{i}\geq l_{j}$$
(6)$$\forall j\in J:\;\sum_{i\in N_{j}}y_{ij}=1$$
(7)$$\forall i\in N_{j}:\;C_{i}^{u}x_{i}\geq \sum_{j\in J}y_{ij}a_{j}^{u}$$
(8)$$\forall i\in N_{j}:\;C_{i}^{d}x_{i}\geq \sum_{j\in J}y_{ij}a_{j}^{d}$$
(9)$$\left(\forall i\in I\right)\left(\forall j\in J\right):\;y_{ij}\leq x_{i}$$
(10)$$\left(\forall i\in I\right)\left(\forall j\in I\right)\left(i\neq j\right):\;d_{ij}\geq d_{\min}x_{i}x_{j}$$
(11)$$\forall i\in E_{f}:\;x_{i}=1$$
(12)$$\forall i\in I:\;x_{i}\in \left\{0,1\right\}$$
(13)$$\left(\forall i\in I\right)\left(\forall j\in J\right):\;y_{ij}\in \left\{0,1\right\},$$
where ε is the additional cost of building a new facility as compared to keeping an existing one. The cost cannot be easily calculated as it depends on many variables, e.g., BS location, difficulty of installation or removal, building, and law processing. The service provider must estimate this parameter for each particular location.

Since constraint (Equation 10) is non-linear, we replace it by the following equation without the product of binary variables $x_{i}$ and $x_{j}$ to obtain a mixed integer programming model.
(14)$$\left(\forall i\in I\right)\left(\forall j\in I\right)\left(i\neq j\right):\;d_{ij}\geq \left(x_{i}+x_{j}-1\right)d_{\min}$$

Constraint (Equation 6) guarantees that the demand *j* is assigned to just one facility at a given moment. All selected facilities must have a sufficient sum of their capacities for uploads and downloads to cover all upload and download demands (in practice, this is an ideal case, network operators are trying to reach that state with the available resources). This is guaranteed by constraints (Equation 7) and (Equation 8). If a facility is selected to be removed from the network infrastructure, none of the demand should be assigned to it, this is given by constraint (Equation 9). To reduce the possible interferences we include the constraint represented by constraint (Equation 10). The $d_{ij},\;i\in I,\;j\in I$ is the distance between centres *i* and *j*. For all UAV pairs the distance will be greater or equal than a certain threshold that can significantly reduce the signal overlaps that usually increases the interferences.

Further, as it is typical in location covering papers where the model enhancements are presented, we provide an alternative in terms of a maximization model. This maximization model considers a predefined number of new facilities (denoted as *p*) to be located with the aim to cover as much as possible, denoted by *p*:

Maximize
(15)$$\sum_{i\in N_{j}}\sum_{j\in J}y_{ij}a_{j}$$
subject to
(16)$$\forall j\in J:\;\sum_{i\in N_{j}}x_{i}\geq l_{j}$$
(17)$$\forall j\in J:\;\sum_{i\in N_{j}}y_{ij}=1$$
(18)$$\forall i\in N_{j}:\;C_{i}^{u}x_{i}\geq \sum_{j\in J}y_{ij}a_{j}^{u}$$
(19)$$\forall i\in N_{j}:\;C_{i}^{d}x_{i}\geq \sum_{j\in J}y_{ij}a_{j}^{d}$$
(20)$$\left(\forall i\in I\right)\left(\forall j\in J\right):\;y_{ij}\leq x_{i}$$
(21)$$\left(\forall i\in I\right)\left(\forall j\in I\right)\left(i\neq j\right):\;d_{ij}\geq \left(x_{i}+x_{j}-1\right)d_{\min}$$
(22)$$\forall i\in E_{f}:\;x_{i}=1$$
(23)$$\sum_{i\notin E_{f}}x_{i}=p$$
(24)$$\forall i\in I:\;x_{i}\in \left\{0,1\right\}$$
(25)$$\left(\forall i\in I\right)\left(\forall j\in J\right):\;y_{ij}\in \left\{0,1\right\}.$$

### 3.2. Model Limitations

The proposed model has several limitations that should be addressed when adopting the model. In the following list of limitations we would like to highlight what should be improved in future research work.

*The model does not modify the FBSs’ configurations*. The model uses the optimal configuration for every single FBS, however, in the final step, the FBS can modify some parameters, e.g., the transmission power to save energy or to optimize spectral efficiency. In this model, we decided not to exceed the real computation complexity of the model, because it would lead to two NP-hard problems in one model. We suggest the adopters of the model to optimize these configurations in the next processing phase. For example, the reinforcement learning techniques can be applied for optimization of the FBS’s parameters to provide a suitable solution.*The model considers one way to reduce the interferences*. In the model, the interferences can be reduced by setting the minimal distance between any two FBSs. However, the model can also include additional ways to reduce the interferences, e.g., to add another objective to find the highest distance between the BS locations;*The model is defined for static scenarios*. If the users unexpectedly change their locations, the current optimal locations have to be re-computed. In practice, it may not present a problem since the data can be prepared beforehand with suitable estimates of user requirements from the particular locations. If necessary, the computation re-run for new requirements is a task that can be run periodically, e.g., every 3, 5, 10 min, according to the requirements.

### 3.3. Model Computational Complexity Considerations

The size of the search space is determined by the number of all possible selections of facilities. For *m* facilities, according to the binomial theorem, it is equal to
(26)m1+m2+m3+⋯+mm=(1+1)m−1=O(2m).

Furthermore, we need to find the most complex condition in extended models for m<n (where *n* is the number of demand areas) to find the resulting computational complexity. In the minimization model these are constraint (Equation 9), and (Equation 13) in the corresponding constraints of the maximization model, which require m·n operations. Based on that the resulting time complexity of these models is $O(2^{m}mn)$.

### 3.4. Designated Implementation

The mathematical model from the Section 3.1, enhancing the LSCP and MCLP models that are originally evolved from SCP, falls into NP−complete class of problems. In this section, we proposed the implementation of a designed model presented in constraints (Equation 4) to Equation (Equation 13) using two heuristic algorithms that ease the integration and reproducibility of the proposed solution into a software solution that may use them.

Since the original model SCP is NP−complete, it is suitable to employ heuristic algorithms to solve such tasks for larger datasets (more than 55-60 UAVs), to reach a solution in a reasonable time. For the UAV deployment use-case of this paper, two promising meta-heuristic algorithms were chosen. First, the CUCKS algorithm with Lévy Flights [43,44], and the DE [15,45], that can provide a suitable solution for this problem.

The CUCKS algorithm pseudocode used to generate a feasible solution (defined by the network designer for a given use-case, e.g., a solution that contains less than *x* UAVs to cover the selected area with the given requirements) is shown in Algorithm 1.
**Algorithm 1** Cuckoo search for UAV deployment pseudocode**Input:** τ terminal condition, e.g., number of iterations 1:**function**cuckooSearchAlgorithm() 2:     Generate initial population of *n* host nests 3:       (particular solution) $x_{i}=1,2,\cdots ,n$ 4:     Evaluate nests $F_{i}=1,2,\cdots ,n$ (using the fitness 5:       function) 6:     check if τ == true;         ▹ if true return the best 7:     **while** τ != true **do** 8:         Get a randomly cuckoo by Levy flights; 9:         Apply repairOperator to get10:            a feasible solution;11:         Evaluate that cuckoo’s quality;12:         Choose a nest among *n*(say,j) randomly13:         **if** $F_{i}>F_{j}$ **then**14:             replace *j* by the new solution;15:         **end if**16:         A fraction (Pa) of worse nests are abandoned17:            and new ones are built;18:         Keep the best solutions;19:         Rank the solutions and find the current best;20:     **end while**21:**end function**

The CUCKS algorithm first generates the initial population of *n* host nests. This, in technical terminology, means that it randomly generates the particular solutions for UAV deployment. The solution is represented by the combination of UAV locations for the deployment. Then, the algorithm evaluates each individual solution by the fitness function. That is defined based on the objective function presented in term (Equation 4) as follows:(27)$$\sum_{i=1}^{n}f\left(x_{i}\right),$$
where *n* is the number of host nests and $x_{i}$ represents the state if the UAV device $x_{i}$ was considered in the solution, 1 if it was and 0 if it was not.
(28)$$\begin{array}{cc} f(x)= & \left\{\begin{array}{c} 1,\;\mathrm{if}\;x=1\\ 0,\;\mathrm{otherwise} \end{array}\right. \end{array}$$

Then, the Algorithm 2 on line six checks if the randomly generated nests in the solution contain the feasible solution. If not, the algorithm works as a standard cuckoo search algorithm with one modification, which is the application of the so-called repairOperator described in more detail below in this section.

Further, we use the DE algorithm to compare the different heuristics to generate a feasible solution. The DE algorithm uses the same objective function as in the case of the cuckoo search algorithm (sum (Equation 27)), however, it does use individuals instead of nests. The DE algorithm first generates the initial population of *n* individuals and in a loop provides the standard flow of DE with a modification of the repairOperator application. This repairOperator is applied before the algorithm inserts the newly generated individuals (after the selection, mutation, and other operations succeeded) into the actual population. The whole DE algorithm to generate a feasible solution is shown in Algorithm 2.
**Algorithm 2** Differential evolution for UAV deployment pseudocode**Input:** τ terminal condition, e.g., the number of iterations; 1:$f_{cost}$: objective function returning the suitability of the newly created solution; 2:specimen: model individual 3:**function**differentialEvolutionAlgorithm() 4:     generate initial population 5:      $P_{0}\leftarrow \left\{x_{1},\;\ldots \;,x_{n}\right\},x_{i}\in {\left\{0,1\right\}}^{n}$; 6:     **while** $\tau :t<t_{max}$ **do** 7:         **for all** $\overset{\to t}{x_{i}}\in P^{t}$ **do** 8:            Generate three random integers, 9:               $r_{1},r_{2},r_{3}\in \left(1,NP\right)$,10:              with $r_{1}\neq r_{2}\neq r_{3}\neq i$;11:            Generate a random integer12:              $j_{rand}\in \left\{1,2,\cdots ,D\right\}$;13:            **for all** *j* **do**14:                  $u_{i,j}^{t+1}\leftarrow$15:                  **if** rand≤CR||j=rand[1,D] **then**16:                      $x_{j,r_{3},t}+F\left(x_{j,r_{1},t-x_{j,r_{2},t}}\right)$;17:                  **else**18:                      $x_{i,j}^{t}$;19:                  **end if**20:            **end for**21:            Apply repairOperator to get22:              a feasible solution;23:            Replace $\overset{\to t}{x_{i}}$ with the child $\overset{\to t+1}{u_{i}}$24:              in the population $\overset{\to t+1}{P}$,25:              if $\overset{\to t+1}{u_{i}}$ is better, otherwise $\overset{\to t}{x_{i}}$ is retained;26:         **end for**27:         t←t+1;28:     **end while**29:            ▹ {$x^{*}$ is the approximation of the optimal solution}30:**end function**

RepairOperator is used in both algorithms. Here, the usage of repairOperator is essential since the solution generated by CUCKS and DE can provide an infeasible solution (a solution that would not cover customers based on the given requirements). For that reason, the repair operator function was designed for modification of the solution provided by both algorithms and changes it to a feasible solution for a given use-case.

The designed repair operator is shown in Algorithm 3. The algorithm sequentially traverses each aspect of the model and adds services to meet that aspect. After going through all aspects of the model (customer availability, capacity, multiple services) it can be said that all the conditions defined by the model are met and the solution to the problem is valid.
**Algorithm 3** Repair operator for both heuristics**Input:**S={1,⋯,m} = the set of all facilities; 1:C={1,⋯,n} = the set of all demands; 2:*I* = the initial matrix containing the covering relationship between *S* and *C*; *M* = the set of demands that requires multiple service (and its value); 3:SC = the set of capacities for facilities; 4:CC = the set of capacities for demands; 5:PS = the possible solution for the problem; 6:CP = the number representing required coverage of the area; 7:**function**repairOperatorAlgorithm() 8:    PSC← get possible solution covering; 9:    **while** givenPercentageIsTrue **do**10:        map(key, value) ← put cover statistics11:         sorted by value;12:          ▹ {Get the statistics about demand covering for each facility in the solution.}13:        PS[i]← get first element from the map;14:        PS[i]← 1;15:        PSC[i]←PSC[i]+1;16:    **end while**17:    **while** multipleServiceSatisfied **do**18:        **for** j←0; *j* < I[j].length; j←j+1 **do**19:           **if** PSC[j]<M[j] **then**20:               **do**21:                   **for** i←0; *i* < *I*.length; i←i+1 **do**22:                       **if** I[i,j]=1 **then**23:                          PS[i]←1;24:                          PSC[j]←PSC[j]+1;25:                          **if** PSC[j]≥M[J] **then**26:                              break;27:                          **end if**28:                       **end if**29:                   **end for**30:               **while** PSC[j]<M[j];31:           **end if**32:        **end for**33:    **end while**34:    **while** capacitiesComparedSatisfied **do**35:        **if** PS capacities > CC capacities **then**36:             PS←37:               add facility with highest capacity;38:        **end if**39:        check if for each demand40:          the capacities are satisfiable using the selected41:          facilities otherwise add additional42:          facility with max capacity that covers43:          particular demand;44:    **end while**45:**end function**    ▹ {The set of facilities (SCS) is now a feasible solution that satisfies capacities and multiple service requirements.}

## 4. Model Verification

The presented model from Section 3.1 is targeted at specific scenarios that are characterized by a high number of users (tens of thousands) spread over large outdoor areas. In other words, scenarios, where multiple UAVs are necessary to cover the area, are expected. The more UAVs are needed the more complex the problem is, and the more beneficial our model becomes. Model definition does not depend on the number of users and UAVs, however the implementation, i.e., the designed algorithms does. To verify that our model is valid and gives realistic and valuable results, it is important to simulate it in a specific scenario. This model was verified on a real outdoor music festival scenario described in Section 4.1. The numerical results are discussed further in Section 4.2.

### 4.1. Scenario Description

The music festival is located on an airstrip of size 2000 by 500 m, however the occupied area by the festival is only 1270 by 400 m with an additional parking area in top left corner of size 70 by 240 m. This considered area of small airstrip is located near Milovice, Czech Republic (GPS coordinates: 50°14${}^{\prime }$10.5${}^{\prime \prime }$ N 14°55${}^{\prime }$15.2${}^{\prime \prime }$ E) which was built in around 1922. In past, it was a military airfield, later it served as an airfield for small aeroclubs. Nowadays, it only hosts a few public events per year and is not used as an airfield anymore, also due to the bad runway condition. The surface is flat and there are no buildings and trees inside the area. This area can host more than one hundred thousand people, but in the past years the usual attendance was between 50 and 80 thousand people per event. The airfield is located in a rural area with a few villages nearby which means that the local telecommunication infrastructure is not prepared for tens of thousands of connections with high throughput demand so the coverage for these events is usually boosted by cells on wheels.

Our scenario estimates 70 thousand attendees. This is a rounded mean value of published numbers of attendees between the years 2007 and 2017. It is the total value of attendees throughout the whole four-day event, this means that the number of attendees at each moment will be lower and will fluctuate. The attendees are divided into seven groups with respect to their capacity requirements. The division was inspired by the example of the traffic model from [46]. The traffic mix was modified according to our estimation in our music festival scenario. It is also estimated, that 50% of users will not be using the internet for various reasons, e.g., sleeping, eating, relaxing, enjoying the festival. The other 50% will be using a mix of traffic shown in Table 3. *Percentage of users* represents the percentage from all participating users during the festival, *Number of users* the percentage of users translated into specific number of users, and *required capacity* the data-rate capacity required by a single user. Aforementioned values were estimated by the authors and are not based on any statistics or studies.

The area is divided into several sub-areas, as shown in Figure 2. Each sub-area has its own purpose, i.e., music, accommodation, parking, common area. The common area works as a connecting point for all other areas, also usually the food stands are located here. According to their purposes, each subarea was assigned a specific user density. Subarea sizes with numbers of users and user densities are shown in Table 4. Users in each subareas are distributed uniformly, 60% of all users are placed into the music area. It is the area where most of the people are gathering. 30% of users are likely in the accommodation and common area, and the last portion of 10% is located in the parking area. For summarizing, the scenario parameters are shown in Table 5.

The FBSs in our scenario play the role of relays. This means that they receive signal from users and transmit it to the internet or the other way round. The wireless equipment can be then divided into two parts, backhaul—connection between FBS and nearby BS, and service—connection between FBS and users. This paper does not concentrate on the backhaul link and it is considered to be ideal and with no throughput limit. The throughput limits are however applied on the service link. There are number of factors that determine the available throughput of the access link, e.g., technology limitations, technology setup—proper antenna alignment, position, and direction, medium limitations—interference.

The FBSs in our scenario are supposed to augment terrestrial BS. However, their design in terms of access link must be slightly modified. The terrestrial BSs expect to serve the users in higher distances, usually higher than 500 m in case of rural area. Additionally, the users are not expected to be located in the close vicinity of the BS (within the radius of one to three hundred meters) and overflowing with thousands of people. However, in case of the FBSs, it is placed directly above the user clusters. For these reasons, the wireless part of the service link must be redesigned. Currently, there are two technologies worth considering for our scenario. FBSs are quite constrained in terms of payload which leads us to consider the IEEE 802.11ac and IEEE 802.11ad, technologies capable of lightweight implementations. The first standard is very well used in many personal and industrial use cases. It is a quite reliable and powerful standard capable of providing a theoretical PHY data-rate of 6933 Mb/s with 4 × 4 MIMO [47]. The second is not so widely utilized but has interesting perspectives with the theoretical PHY data-rate of 6756.75 Mb/s on a single stream [48]. These values are however not realistic in live scenarios. Not all available devices are capable of providing such high MIMO capabilities and it is not likely that the channel conditions will be good in distances over 50 m. In addition to that, current available devices are usually limited on the other side by 1 Gb/s Ethernet cable. Additionally, it is important to note that PHY data-rate is not what the end user receives and the data-rate on the application layer is more suitable. It is estimated that the usual drop between physical and application layer can be typically between 20% and 30%, adding another 20 to 30% loss due to the channel conditions, then the more realistic data-rate on the service side would be around 3 Gb/s. This value was selected as the service link limit for each FBS.

There are two cell towers in the nearest area, both of them with LTE capabilities. The two towers combined provide coverage for all three operators in the Czech Republic. According to [49], typical data-rates for each LTE BS is 1 Gb/s uplink and 500 Mb/s downlink. This would mean that the total of 4.5 Gb/s could be theoretically available from the local infrastructure. The total data-rate requirements for all users in our scenario is 29,610 Mb/s. The local cell towers could then cover roughly 15% of the demand. However, since there is also a demand from local villages, it is not possible to be sure of the cell towers availability and for this reason the local infrastructure is neglected from the calculations. The data about cell towers were obtained from [50].

### 4.2. Numerical Results and Discussions

For a demonstration, two datasets (C and G) are visualized in this section. The original distribution of FBSs for dataset C is shown in Figure 3 with the results shown in Figure 4 and Figure 5. In case of dataset G, the original distribution of FBSs is shown in Figure 6 with the corresponding results in Figure 7 and Figure 8. Small blue dots symbolize users, different densities are clearly visible. Red crosses symbolize potential locations for placing FBSs. Red circles show the radii of each FBS. The radii are not visualized in Figure 3 and Figure 6 for the picture to stay clear. The whole process of data generation and processing consisted of three major steps. First, the two original distributions of FBS were generated, as well as the distribution of users. Each FBS and each user has unique coordinates in the area. Second, the data from the first step were passed to the two algorithms for calculation. Third, the output data from the algorithms were visualized by the same code as in the first step. MATLAB tool was used in steps one and three. Step two was implemented in OpenJDK without any specific external library.

The number of possible solutions is high and a reasonably good solution had to be selected. The input data consisted of eight datasets named by capital letters from A to H. A total of sixteen simulations were run, eight simulations per each algorithm. The results were selected based on the number of FBSs needed for the deployment, in our case the smaller number the better. When two results had the same number of FBSs, the solution taking less computation time was selected. The computation was performed on a personal computer with parameters described in Table 6. Ten thousand iterations were performed for each algorithm. In these runs, the resulting number of FBSs varied from the best result to plus one or two more locations. The results are shown in Table 7.

Two cases of FBS deployment were investigated, one where FBSs are distributed evenly in a grid inside the whole area and second where the FBSs are also distributed evenly but on the edges of the area. The second deployment was investigated for cases when FBSs are not allowed to fly over crowded areas for safety reasons. The original distribution of the first case for dataset C is shown in Figure 3, the FBSs are deployed in a grid with distances of 100 m. This lead to the total number of 70 initial deployment locations. The initial locations of the second case for dataset G are shown in Figure 6. Here, the distances between FBSs were shortened to 50 m to reach similar number of original locations as in case one. The number of initial locations plays a big role in calculation time of the minimized solution. According to the results from Table 7, it was proven that numbers between 60 and 70 symbolize a certain threshold for calculation complexity, because for the datasets D and H where the numbers of initial locations were 90 and 85 prolonged the calculation time radically. Longer time would compromise the usefulness of the algorithms for repeated use in short periods of time, for instance in case of high user mobility. On the other hand, it would make the result more accurate.

The processing of the algorithms takes certain amount of time. Specifically for datasets C and G, processing of the CUCKS took 2274 s for the case with the BS inside of the area and 1856 s for the BS outside of the area. These values are approximately four to six times higher than those for DE: 373 s for the case with the BS inside of the area and 387 s for the BS outside of the area. The number of resulting locations in all four cases was ten. The reason for this was likely that the required average data-rate of all users combined at each moment was 29,610 Mb/s. If this value is divided by the available throughput on each FBS (3 Gb/s), we achieve ten. This means that the resulting number of locations is highly affected by the throughput limit of the FBSs rather than by the radius of their wireless devices.

The distribution of FBSs seems logical for both algorithms and both cases. The algorithms validate the expected result that more locations will be selected over areas with higher user densities. It is visible more in the cases with locations inside of the area, i.e., Figure 4 and Figure 5. Clearly, the user density forces the algorithms to select more locations in the denser areas. Both algorithms give similar results in terms of FBS distribution, however if the preference was the calculation time, the DE should be the favorable option. The shorter calculation time would be appreciated in scenarios where the FBSs’ position would be constantly updated. The reason why DE is so much faster than CUCKS is that the DE algorithm is not generating the new pool of solutions for each generation in opposite to the CUCKS algorithm.

The data-rate and radius parameters used in the scenario were derived from theoretical capabilities of two technologies—IEEE 802.11ac and IEEE 802.11ad. Even though the results support the theoretical values, it is difficult to predict the outcome of a real implementation because there are still a great number of variables during live test, e.g., weather conditions, inter-BS and inter-user interference, and rapid user mobility. That means, real measurements are required to support the theoretical values. Even though the measurements would show much lower performance, future of mmWave communications might provide even greater and more stable parameters with the new IEEE 802.11ay standard. It is an updated IEEE 802.11ad standard promising extended range and higher throughput provided by newly implemented multiple-input and multiple-output (MIMO) feature.

Let us now discuss the limitations of used methods in this research. The limitations lie in the initial number of FBSs locations, since increasing this number would extend dramatically the calculation time. As it was established for our case, this would not be such a critical issue because of the high number of users in the area. In some other cases however, it might be crucial to position the FBSs into more precise spots. Further, in our scenario, there is theoretically unlimited number of FBSs available to cover the area. If the user requirements would rise or if the FBS capabilities would be lower, the number of FBSs could rise to the point where the total implementation price would make the solution unfeasible. This condition can be implemented in the algorithm, however it is not for the reason that authors wanted to keep the decision making under their control.

For future research directions we expect to implement and verify more heuristic algorithms and also provide additional models targeting a situation in which there is a specific number of drones (significantly limited resources). The focus will be more on how to cover as most as possible locations using the optimized heuristic parameters, with a detailed focus on scenarios where the users are quickly moving to different locations.

One of the important factors to be considered is both inter-BS and inter-user interference. Including interference reduction into described algorithms would radically increase complexity of the solution in the way that it would become even more difficult to obtain a solution with regards to calculation time and computational complexity. One option to address the interferences problem is in the next stage after obtaining the solution from the algorithms. This is planned as the next step in this research. There are various approaches to address the interference problem from which the following approaches are planned by the authors:
Setting minimum distance from one FBS to another;Using mathematical model with multi-objective function–minimize number of FBSs and maximize distance between BSs;Reducing radius of the FBSs by reducing antenna gain;Using different radio frequencies among neighboring FBSs.


## 5. Conclusions

The key objective of this paper was the optimal FBS distribution over a desired area while minimizing the number of necessary FBSs. Two novel mathematical models were designed for this purpose. The models take into account several aspects that are crucial in the presented scenario. It includes the capacities for both downlink and uplink, the consideration of existing base station nodes in the UAV deployment area, and the possibility to select lower priority locations that may be excluded from the coverage. Further, we consider the computational complexity of these models for very large datasets, which are defined by tens of thousands of users and tens of FBSs. We employ cuckoo search and differential evolution algorithms with the developed repairOperator providing a feasible solution to this problem.

To verify the viability of the models, data from a real-life scenario were used. The Presented scenario was a crowded music festival with various user densities. Two use-cases for FBS distribution were tested. First with FBSs distributed inside the area and second with FBSs outside the area. The two algorithms were implemented and the optimal solution was obtained via extensive simulations with eight datasets. Both of the algorithms were able to find a solution. Major difference was in the computation time where the CUCKS obtained results in time three to six times higher for the two use-cases than the DE. It is however important to consider that even though the higher number of candidate locations can provide higher accuracy of the final solution, it can lead to unnecessary waiting time. It is recommended for future model adopters to find an optimal number of the candidate locations suitable for their specific use-case.

The data used as inputs for the implemented algorithms were mostly based on theoretical values, and the research would benefit greatly from support of real-life measurements. The measurements might reveal some deficiency in data-rates and FBSs’ radii. One cause of the deficiency might be signal interferences, that have not been addressed in this research. However, as discussed in Section 4.2, including interferences might make our models excessively complex, hence the issue of interferences will be addressed in detail in our future research.

## Figures and Tables

**Figure 1 sensors-21-05580-f001:**
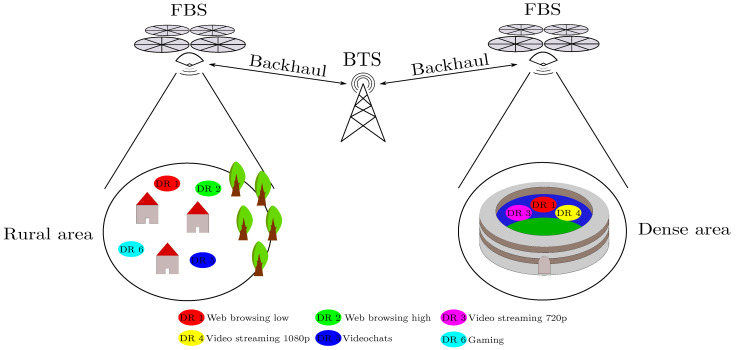
Use-cases with FBSs assisting current infrastructure to extend the coverage in the area.

**Figure 2 sensors-21-05580-f002:**
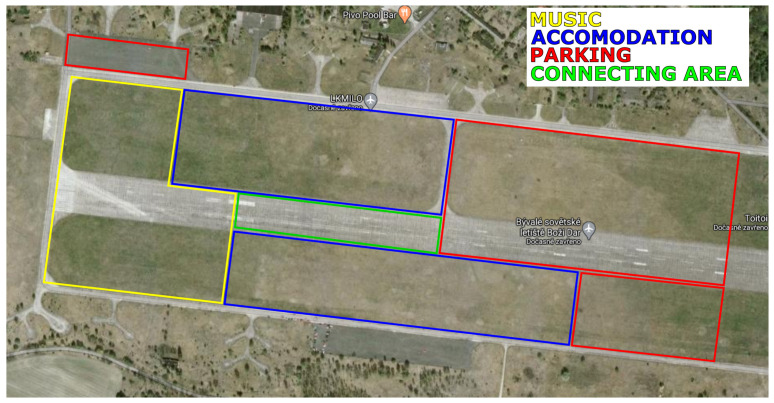
Real satellite photo of the airfield with marked subareas.

**Figure 3 sensors-21-05580-f003:**
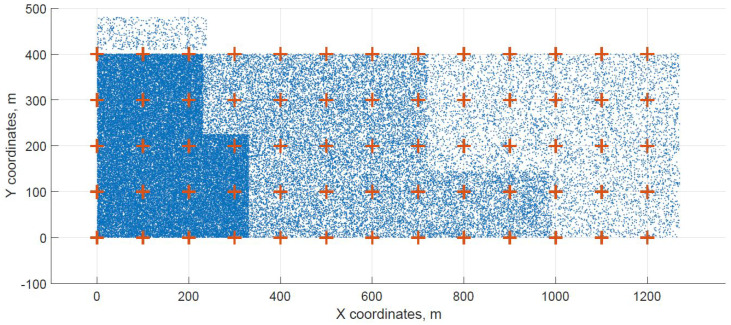
Original distribution grid with FBSs placed inside of the area for dataset C.

**Figure 4 sensors-21-05580-f004:**
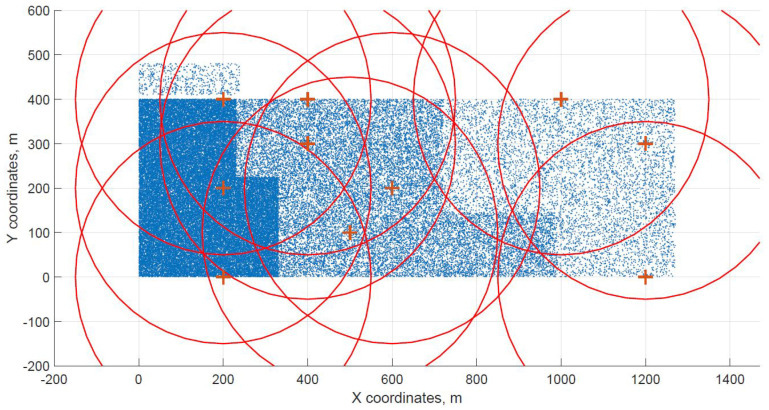
Cuckoo search resulting grid for dataset C.

**Figure 5 sensors-21-05580-f005:**
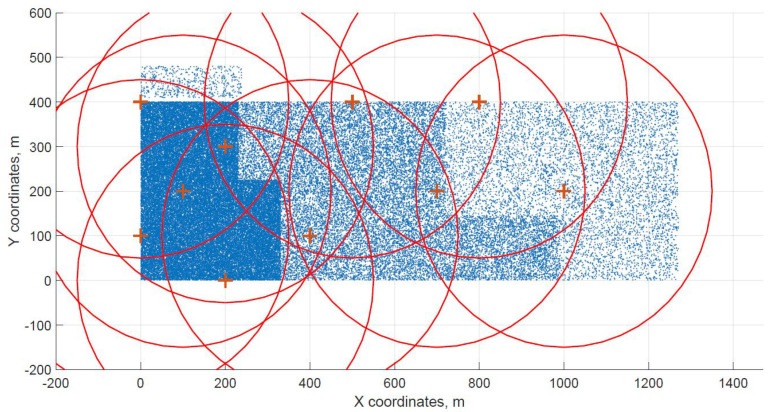
Differential evolution resulting grid for dataset C.

**Figure 6 sensors-21-05580-f006:**
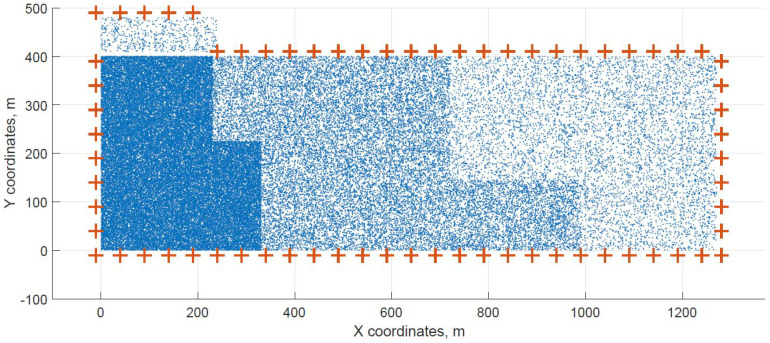
Original distribution with users outside of the area for dataset G.

**Figure 7 sensors-21-05580-f007:**
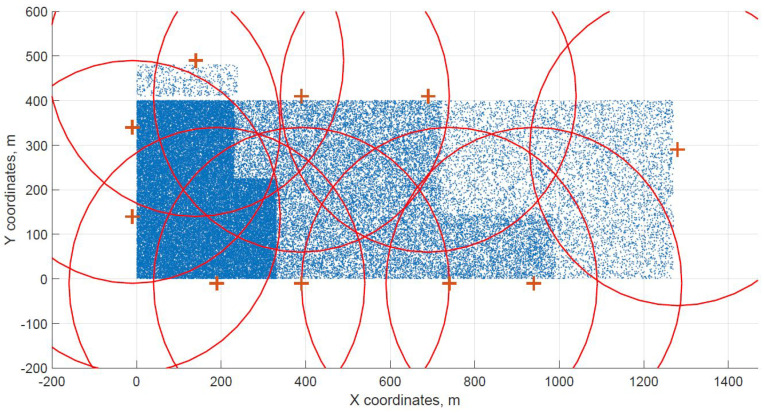
Cuckoo search resulting grid for dataset G.

**Figure 8 sensors-21-05580-f008:**
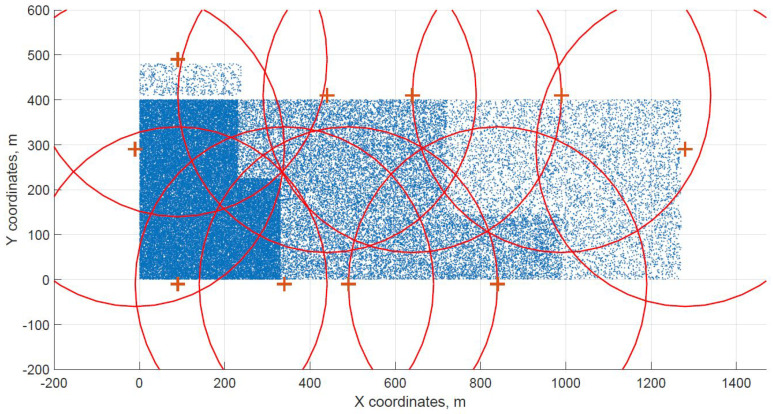
Differential evolution resulting grid for dataset G.

**Table 1 sensors-21-05580-t001:** Summary of the most recent papers on location optimization problem in FBSs use-cases.

Used Algorithms	Use-Case	Objective	Published
Multi-Population GA for horizontal dimensions placement, Mixed Integer Second Order Cone Problem for altitude placement.	Congested area containing a set of users. The terrestrial Base Station (BS) cannot provide service to users. A UAV BS is deployed in order to provide service to as many users as possible. The users have different Quality of Service (QoS) requirements.	Max. no. of covered UEs with different QoS.	2018 [27]
Novel alg.: Adaptive Multiple drone base Station placement.	UAV BS serve as relays in hotspot area to assist to the macro BS	Min. no. of UAVs and satisfy the QoS of UEs.	2018 [22]
Geometric relaxation, K-means deployment, Power efficient K-means deployment, Robust Deployment with imperfect user location information.	Terrestrial infrastructure is unavailable. Required support from UAV BS.	Max. no. of covered UEs.	2018 [28]
Centralized deployment algorithm, distributed motion control algorithm.	UEs are distributed randomly and in clusters, also, static and dynamic scenarios are considered. Two environments – with and without obstacles. Two different initial states for FBSs.	Min. number of UAVs, cover all UEs. Max. no. of covered UEs.	2018 [26]
Novel alg. based on GA.	Real environment with different UE densities.	Max. no. of covered UEs.	2019 [29]
Novel alg.: Edge-prior.	Random user distribution with known positions.	Min. number of UAVs, cover all UEs.	2019 [23]
Novel alg. based on GA.	Existing deployment of static base stations	Max. UE throughput and min. consumption.	2019 [31]
Hybrid alg.: Centralized greedy search alg. for determining the no. of FBSs. Distributed motion alg. for enabling each FBS to autonomously control its motion toward the optimal position.	UAVs with or without the support of ground BS. Distribution of UEs is unknown.	Min. no. of UAVs, max. load balance.	2019 [24]
UAV-artificial bee colony.	Deployment of UAV BS in post disaster scenario.	Max. network throughput.	2019 [32]
Novel alg.	mmWave network, serving all ground users, predefined set of locations.	Min. number of UAVs, cover all UEs.	2020 [25]
K-means clustering and stable marriage approach to find 2D positions. Space constrained exhaustive search and PSO to find the optimal altitudes of the FBSs.	UEs are distributed with homogenous Poisson point process. When a ground station is damaged and stops transmitting, UAVs are deployed in the area with lost connectivity.	Max. spectral efficiency, maintain QoS.	2020 [33]
Sequential Exhaustive Search, Sequential Maximal Weighted Area.	Target area with two sets of users demanding either the same or different QoS requirements.	Max. no. of covered UEs with the same and different QoS.	2021 [30]

**Table 2 sensors-21-05580-t002:** Mapping mathematical terminology to communication networks terminology.

Mathematical Terminology	Wireless Networks Terminology
Facility	UAV or base station node
Demand	A user in a given area
Capacity	Throughput that is requested by sum of user requirements in a given area to cover
Multiple service	A user requires to be potentially covered by the *x* UAV or base station nodes.
Existing service	Usually base station nodes that already exists in the area to cover and should remain after the reconfiguration or deployment phase

**Table 3 sensors-21-05580-t003:** Traffic mix distribution among users.

	Percentage of Users [%]	Number of Users [-]	Required Capacity [Mb/s]
No connection	50	35,000	0
Web browsing low	30	21,000	0.2
Web browsing high	10	7000	0.8
Video streaming 720p	3	2100	3.5
Video streaming 1080p	2	1400	6
Videochats	3	2100	2
Gaming	2	1400	3

**Table 4 sensors-21-05580-t004:** Subarea division with user densities.

	Music Area	Accommodation and Common Areas	Parking Area
Percentage of users from total number	60%	30%	10%
Number of users	42,000	21,000	7000
Section, square meters	114,500	212,650	197,650
User density, user/square meter	0.3668	0.0987	0.0354

**Table 5 sensors-21-05580-t005:** Scenario parameters.

Parameter	Value
Area size	1270 × 400 m
Number of users	70,000
Users’ data-rate requirements	See Table 3
FBSs’ coverage radius	350 m
FBSs’ max throughput	1 Gb/s

**Table 6 sensors-21-05580-t006:** Testing environment parameters.

OS	System Type	CPU	RAM
Windows 10 PRO	64-bit Operating System, x64-based processor	Intel(R) Core(TM) i7-7700 CPU @ 3.60 GHz 3.60 GHz	16.0 GB

**Table 7 sensors-21-05580-t007:** Calculation results for CUCKS and DE algorithms using 10,000 iterations.

	Theor. No. of Candidate Locations to Deploy UAVs	CS	DE	CS	DE	Dataset
		Number of FBS		Calc. Time, s		
FBS generated inside of the area	30	10	10	1019	329	A
	50	10	10	1850	358	B
	70	10	10	2274	373	C
	90	10	10	3156	554	D
FBS generated outside of the area	25	10	10	918	302	E
	45	10	10	1530	369	F
	65	10	10	1856	387	G
	85	10	10	2844	523	H

## Data Availability

Not applicable.

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
