# Peer review of "Modeling Optimal Location Distribution for Deployment of Flying Base Stations as On-Demand Connectivity Enablers in Real-World Scenarios"

_sensors, 2021, doi:10.3390/s21165580_

Round 1
Reviewer 1 Report
The work of this paper is practical and logical. However, there are some problems to be further improved as well:
1.Authors should point limitations of the model.
2.Another obvious problem with this paper is the lack of enough experimentation to demonstrate the validity and applicability of the proposed method. The author needs to do more experiments and show them in this paper.
3.Authors should consider comparing and citing the following work. "Situational Assessment for Intelligent Vehicles based on Stochastic Model and Gaussian Distributions in Typical Traffic Scenarios".
Reviewer 2 Report
In this paper, the authors consider the location issue in traffic network. This problem can be changed to a coverage problem. The authors use a mathematical model to represent the deployment of flying base station, especially in large-scale scenarios. In fact, this model is represented by an optimization algorithm. The authors consider a location set and minimize the number of flying base stations by finding the optimal locations. Also, the authors consider the constraints. For search optimization, the authgors propose heuristic search method. Finally, simulation result is given to illustrate the effectiveness of the proposed algorithms.
Theoretically speaking, the authors change the problem into an optimization algorithm. This is convenient for computing handling. It is also clear for readers to understand the problem and solve it.
Technically speaking, the optimization algorithm may use several existing methods to solve it. In fact, it is a mixed integer programming (MIP). This problem has been used in allocation tasks issue.
The authors may revise the paper when considering the following points.
- In the introduction section, the authors may mention the current issue is one type of coverage problem. The authors may cite the following paper " Coverage control of multiple unmanned aerial vehicles: A short review" S Huang, RSH Teo, WWL Leong, N Martinel, GL Forest, C Micheloni Unmanned Systems 6 (02), 131-144,2018
- In the optimization (14), it is a standard MIP problem. The authors may give some comments regarding this point.
- In the section 4, I was impressed by your model validation. It is better if you can provide a comparison with existing one.
